# Modulating Activity Evaluation of Gut Microbiota with Versatile Toluquinol

**DOI:** 10.3390/ijms231810700

**Published:** 2022-09-14

**Authors:** Long-Long Zhang, Ya-Jun Liu, Yong-Hong Chen, Zhuang Wu, Bo-Ran Liu, Qian-Yi Cheng, Ke-Qin Zhang, Xue-Mei Niu

**Affiliations:** State Key Laboratory for Conservation and Utilization of Bio-Resources, Key Laboratory for Microbial Resources of the Ministry of Education, School of life Sciences, Yunnan University, Kunming 650091, China

**Keywords:** *Enterococcus faecalis*, *Enterococcus faecium*, toluquinol, acetylation, gut microbiota, biotransformation, antitumor

## Abstract

Gut microbiota have important implications for health by affecting the metabolism of diet and drugs. However, the specific microbial mediators and their mechanisms in modulating specific key intermediate metabolites from fungal origins still remain largely unclear. Toluquinol, as a key versatile precursor metabolite, is commonly distributed in many fungi, including *Penicillium* species and their strains for food production. The common 17 gut microbes were cultivated and fed with and without toluquinol. Metabolic analysis revealed that four strains, including the predominant *Enterococcus* species, could metabolize toluquinol and produce different metabolites. Chemical investigation on large-scale cultures led to isolation of four targeted metabolites and their structures were characterized with NMR, MS, and X-ray diffraction analysis, as four toluquinol derivatives (**1**–**4**) through O1/O4-acetyl and C5/C6-methylsulfonyl substitutions, respectively. The four metabolites were first synthesized in living organisms. Further experiments suggested that the rare methylsulfonyl groups in **3**–**4** were donated from solvent DMSO through Fenton’s reaction. Metabolite **1** displayed the strongest inhibitory effect on cancer cells A549, A2780, and G401 with IC_50_ values at 0.224, 0.204, and 0.597 μM, respectively, while metabolite **3** displayed no effect. Our results suggest that the dominant *Enterococcus* species could modulate potential precursors of fungal origin and change their biological activity.

## 1. Introduction

Increasing evidence suggests that gut microbiota play a crucial role in human health, being involved in the digestion of food, in the protection of mucosal surfaces, and in modulation of the immune system [1,2]. The health benefits and detrimental consequences that the interactions between dietary and microbial factors elicit in the host have become a fascinating yet largely unexplored facet of diet–microbiome–host relationships [1,2]. Most studies focused on the effects of gut microbiota as a whole on transforming and degrading the common metabolites in diet, such as catabolizing phenylalanine into indole and phenylacetate [3]. Moreover, the modulation of common bioactive metabolites mainly from plants, pharmaceuticals, and other environmental chemicals in gut intestines have also received immense interest [4,5]. However, what types of specific gut microbial strains and how they function in modulating specific intermediate metabolites in diet still remain largely unclear.

Fungal metabolites are present throughout the food chain and occur in raw crops and crop by-products [6,7,8,9]. Most studies reported the effects of gut microbes on maintaining intestinal homeostasis via biotransformation of mycotoxins into less toxic metabolites and the compromised intestinal barrier mechanism induced by mycotoxins. Several well-known mycotoxins, including aflatoxins, citrinin, deoxynivalenol, fumonisins, nivalenol, ochratoxin A, patulin, and zearalenone, and their effects on gut-microbiota-mediated host responses have been investigated [7,8,9]. However, studies on the role of specific gut microbial strains in modulating specific intermediate metabolites from fungal origins in diet are still lacking. 

Previous studies revealed that a polyketide synthase gene cluster (PKS) containing a 6-methyl salicylic acid synthase gene was responsible for the biosynthesis of the famous patulin [10,11,12]. It has been elucidated that 6-methyl salicylic acid, *m*-cresol, and toluquinol are the first, second, and third precursors, respectively (Figure 1A) [12]. Among these precursors, toluquinol was also reported as a naturally occurring metabolite and firstly isolated from the fungus *Penicillium* sp. [13,14]. Toluquinol is also a key precursor for biosynthesis of a unique class of PKS-TPS hybrid prenyl epoxy-cyclohexenoids (PECs), which are widely distributed in plant pathogenic fungi, including the potent antifungal yanuthones from *Aspergillus* sp. and the notorious phytotoxic macrophorins from the fungus *Macrophoma* causing fruit rot of apples (Figure 1B) [15,16]. 

Our previous studies revealed that the predominant nematode-trapping fungus *Arthrobotrys oligospora*, a ‘biological indicator’ of nematodes in mushroom growth facilities and an ideal natural agent against parasite nematodes [17,18], also harbored toluquinol-derived PEC metabolites, athrobotrinsins, and arthrosporols (Figure 1B) [19,20]. These autoregulators contribute to fungal diverse antagonistic action and play a key role in fungal colonization and nematicidal trap formation [21,22]. Noteworthily, toluquinol was a dominant metabolite in one *A. oligospora* mutant ∆276 [23]. Moreover, previous studies have reported that toluquinol exhibits a variety of biological activities, including antitumor activity, and has been assumed as a promising drug candidate for the treatment of cancer [13,14]. In particular, our recent study suggested that the common gut microbe *Escherichia coli* could use toluquinol as a building block to produce more effective antitumor fluorescent arthrocolins with a unprecedented carbon skeleton [24]. Thus, we consider the question, are any other gut microbe that could also use toluquinol as substrate and how?

In this study, we first analyzed the distribution of the biosynthetic pathway for toluquinol biosynthesis in 2180 published fungal genome sequences; then, we performed metabolic investigation on the food and fruit with *Penicillium* colonization for the association of toluquinol with *Penicillium* species. Chinese steamed bread is the staple carbohydrate of the northern Chinese diet, and tangerine orange is a popular citrus fruit in southern China [25,26]. It is a general phenomenon that these food and fruits stored in a damp environment might be dotted with a typical color diagnostic for *Penicillum* sp. So, these two materials were applied for *Penicillum* sp. colonization and toluquinol existence. Then, 17 common strains of human intestinal probiotics belonging to 12 genera [27] were fermented and fed with toluquinol. All the 17 strains fed with solvent only were used as negative controls. The metabolic profiles of the extracts of all the 34 gut microbial samples were analyzed with HPLC-MS/PDA. The dominant *Enterococcus* genus, *Enterococcus faecalis* (*E. faecalis*) and *Enterococcus faecium* (*E. faecium*), could modulate toluquinol and yield four distinct metabolites. Large-scale liquid cultural fermentation combined with chemical investigation led to isolation of these targeted compounds. The structures of the metabolites were determined with HRMS and NMR, and further with X-ray analysis. Interestingly, metabolites with a rare methylsulfonyl group were found in four strains including the *Enterococcus* species. Then, four strains were further fermented and fed with less active toluquinol analogue hydroquinone and analyzed. Moreover, these toluquinol-derived compounds were evaluated for their biological activities towards three different cancer cells.

## 2. Results and Discussion

### 2.1. The PKS Gene for Toluquinol Biosynthesis Was Widely Distributed in Fungi and Predominant in Penicillium Species

Analysis of 2180 published genome sequences of fungi belonging to 1844 fungal species in 818 fungal genera revealed that the toluquinol biosynthetic pathway, such as the known Mac A-Mac C., existed in 289 strains belonging to 246 species in 100 genera (Figure 1C). In particular, analysis of 45 published genome sequences of fungi *Penicillium* genus in 39 fungal strains revealed that the toluquinol biosynthetic pathway existed in 22 strains belonging to 17 fungal species, including the species *P. roqueforti*, for production of blue cheeses (Figure 1C) [28]. *Penicillium* species are the most common and versatile fungi that play important roles in the processing of feed and food products, including mold-fermented meat sausages, dry-cured meat production, and pigmented liquid fermentations [29]. *Penicillium* species, especially *P. roqueforti*, is often applied in the production of well-known valuable blue cheeses [28,29]. 

### 2.2. Toluquinol as a Natural Metabolite in Food and Fruits with Penicillium Species 

In order to evaluate if toluquinol was produced in common *Penicillium* species, a fungal strain was collected and purified from Chinese steamed bread stored in a fridge for two weeks and infected with cyan *Penicillium*
*chrysogenum* species (Figure 1D). The fungal strain was inoculated on some Chinese steamed breads. After 5 days, these Chinese steamed breads were fully covered with the cyan fungus. Comparison of HPLC profiles of the ethyl acetate extracts revealed that the Chinese steamed breads with cyan mold had more peaks than Chinese steamed breads without cyan mold (Figure 1D). Among them, one peak was assigned to toluquinol for the identical Rt at 7.8 min and UV absorption to the standard sample. 

Inoculation of the fungus *P. chrysogenum* in some tangerines was performed in a similar way. After only 4 days, these tangerines were fully covered with the fungus (Figure 1E). Similarly, HPLC profiles of the ethyl acetate extracts revealed that toluquinol was detected in the tangerines with the fungus *P. chrysogenum*, but not in tangerines without the fungus (Figure 1E).

### 2.3. Evaluation of the Modulating Activity of 17 Gut Microbes with Toluquinol

Metabolic profiles of the ethyl acetate extracts of all the strains fed with and without toluquinol revealed that two species belonging to *Enterococcus* genus, *E. faecalis* and *E. faecium*, and another two strains, *Stretpococcus thermophiles* and *Lactococcus lactis* subsp., were capable consuming toluquinol and yielded extra and distinct peaks in the HPLC spectra (Figure 2A,D and Appendix A). Among them, two *Enterococcus* species shared three quite similar targeted peaks at the same retention times (Rts) 19.3, 17.8, and 9.6 min, and *Stretpococcus thermophiles* and *Lactococcus lactis* subsp. shared two quite similar targeted peaks at the same Rts of 12.1 and 9.6 min. Interestingly, all the four strains shared a quite similar extra peak at the same Rt of 10 min. Further analysis of all the targeted peaks showed that peaks **1** and **2** shared the similar UV absorptions to toluquinol, at 194 nm and 278 nm while peaks **3** and **4** displayed dramatically increased UV absorptions with 199/307 nm for **3** and 210/308 nm for **4** (Figure 2E–H). High-resolution mass spectrometry showed that peaks **1** and **2** shared the same molecular formula based on their negative ion peaks at *m/z* 165.05479 for **1** and 165.05444 for **2**, while peaks **3** and **4** shared the same molecular formula based on 201.02170 for **3** and 201.02184 for **4** (Figure 2I–L). These results indicated that two *Enterococcus* species, *E. faecalis* and *E. faecium*, could transform toluquinol via two different substitution reactions, while *Stretpococcus thermophiles* and *Lactococcus lactis* subsp. did via only one substitution reaction.

### 2.4. Characterization of Toluquinol-Derived Metabolites in E. faecalis and E. faecium 

*E. faecalis* was cultivated in a 50 L culture of LB medium fed with toluquinol. Chemical investigation on the ethyl acetate extract of the liquid culture led to isolation of the three targeted metabolites **1**–**3**. Analysis of ^1^H NMR spectra of metabolites **1** and **2** displayed that both **1** and **2** had one more acetyl group than toluquinol (Appendix A), consistent with their molecular formula revealed with the MS. A comparison of their ^1^H NMR spectra with those reported in the literature [30] revealed that the acetyl group was attached to 4-OH in **1** and 1-OH in **2**, respectively, consistent with their Rts in the HPLC profile. Different from the characteristic 1,2,4-trisubstituted pattern in ^1^H NMR spectra of compounds **1** and **2**, compound **3** displayed only two single protons at 7.06 and 6.65 ppm and two methyl single peaks at 3.10 and 2.08 ppm, which suggested that one more group with one methyl group was attached to C-5 of toluquinol. According to the MS and NMR data between toluquinol and **3** (Appendix A), there was one extra methylsulfonyl group in **3**. In order to clarify the rare methylsulfonyl group in **3**, X–ray diffraction was performed, and the structure of **3** was finally determined as 5-methylsulfonyl toluquinol (CCDC 2181025). Compound **4** was established as 6-methylsulfonyl toluquinol by comparison of the MS and UV data of **4** with those of the known product in the reference [31]. It was interesting to note that *Enterococcus* species could transfer either an acetyl group to the OHs or a methylsulfonyl group to C-5 of toluquinol. Acetylation is a general metabolic reaction common in organisms [32]. It is reasonable that *Enterococcus* species might consume endogenous acetyl group for the acetylation of toluquinol, which is also consistent with the fact that *Enterococcus* species were lactic acid bacteria (LAB) [33]. 

A previous study suggested that in the presence of H_2_O_2_ and Fe^2+^, DMSO could be broken into a methyl radical and a methylsulfonyl group via Fenton’s reaction (Figure 3), which could turn benzonquinones, such as hydroquinone (**5**), into majorly methylated ones (**6**–**8**) and minorly methylsulfonylated benzonquinols (**9**–**11**) [31]. Thus, we hypothesized that the methylsulfonyl group might be donated by solvent DMSO also via Fenton’s reaction in *E. faecali*. Toluquinol dissolved in methanol was also used as a substrate and added into the fermentation culture of *E. faecali*. Metabolic profiles of the extracts of *E. faecali* fed with toluquinol in methanol revealed that metabolite **3** disappeared in the culture with methanol, while **1** and **2** still existed the same in the culture with DMSO (Figure 4). It is noteworthy that the methylsulfonyl group only existed in the benzonquinol derivatives (**9**–**11**), and not in the benzonquinone derivatives (**6**–**8**), consistent with our results that **3** and **4** were both toluquinol-derived metabolites with the methylsulfonyl group, indicating that the methylsulfonyl groups in **3** and **4** were also derived from DMSO via Fenton’s reaction. Interestingly, no toluqinone (namely, oxidized toluquinol) or its derived metabolites were detected in the metabolic profiles of the extract of *E. faecali*.

Further, an experiment where hydroquinone (**5**) dissolved in DMSO was also applied as a substrate and added into the fermentation culture of *E. faecali* was carried out. Interestingly, no extra peaks were observed in *E. faecali* cultures fed with hydroquinone (**5**) compared with those fed without hydroquinone (**5**) (Figure 5). Neither did the other 15 strains, including *E. faecium*, *Stretpococcus thermophiles*, and *Lactococcus lactis* subsp. fed with hydroquinone (**5**) (Figure 5 and Appendix A), suggesting that distinct from toluquinol, hydroquinone (**5**) could not induce microbial modulation among these gut microbes. This might be ascribed to having one more methyl group in toluquinol compared with hydroquinone (**5**), which might add the specialties of instability and the environmental sensitivity of toluquinol. Previous studies suggested that brown rot fungi exploit the Fenton reaction to depolymerize and degrade biomass [34]. Here, we found that the predominant *E. faecali* and *E. faecium*, together with other two strains *Stretpococcus thermophiles* and *Lactococcus lactis* subsp., could also use Fenton’s reaction to modulate the versatile precursor toluquinol. It is reported for the first time that metabolites with the methylsulfonyl group are biosynthesized in organisms.

### 2.5. Effect of the Structural Modification on the Biological Activity

Previous studies reported that toluquinol had antitumor activity [13,14]. Thus, we evaluated the antitumor activities of these three toluquinol-derived metabolites (Figure 6, Table 1). Similarly, **1** and **2** also displayed multiple antitumor activities against human non-small-cell lung carcinoma cell A549, human melanoma cell A375, human ovarian cancer cell A2780, human Wilms tumor cell G401, colorectal cancer cell HCT116, and hepatocellular carcinoma cell LM3. Among them, metabolites **1** and **2** displayed dramatically stronger inhibition towards cancer cell line A549 but obviously weaker inhibition towards LM3 than toluquinol, suggesting that the acetylation at OH group could modulate the biological activity. Notably, **1** displayed the strongest inhibitory effect on A549, A2780, and G401 with IC_50_ values of 0.224, 0.204, and 0.597 μM, respectively (Figure 6, Table 2). Particularly, **1** showed 21.3 times stronger inhibition against A549 than toluquinol. However, **3** displayed no antitumor activities in the tested cancerous cell lines, suggesting that the substitution at phenyl ring could result in the absence of antitumor activities. A previous study suggested that toluquinol could alter endothelial cell cycle distribution and induce apoptosis [13,14]. Our detailed analysis showed that **1** and **2** were quite similar to toluquinol in inducing apoptosis rather than necrosis in A549, but the apoptosis induced by **1** or **2** were quite stronger than toluquinol via arresting the cell cycle of A549 at G2/M phase (Figure 7). Our results indicated that toluquinol-derived metabolites **1** and **2** could be potent anticancer agents towards A549.

Among 17 strains belonging to 12 genera, two strains—*E. faecalis* and *E. faecium*—together with other two strains—*Stretpococcus thermophiles* and *Lactococcus lactis* subsp. —displayed strong metabolic modification ability towards exogenous small molecule toluquinol via acetylation and/or methylsulfonylation. *E. faecalis* and *E. faecium* are among the most predominant species in the gastrointestinal tracts of animals and humans [35]. Interestingly, *E. faecalis* is more generally distributed in the intestines of humans, whereas *E. faecium* is widely associated with humans and other animals [35]. Importantly, both strains are widely used in the production of fermented dairy foods and contribute to the sensory, nutritional, and safety properties of fermented food products such as cheeses and sausages through secreting lactic acid, fatty acids, vitamins, antibiotics, etc. [36,37]. Moreover, *E. faecium* and *E. faecalis* could offer useful protection against unwanted bacteria in food technology via producing antibacterial proteins (bacteriocins) that are inhibitory against food spoilage or pathogenic bacteria, such as *Listeria monocytogenes, Staphylococcus aureus, Vibrio cholerae, Clostridium* spp., and *Bacillus* spp. [36,37,38]. 

Similarly, in production animals such as poultry, cattle, and pig, *E. faecium* is the prevalent species [39]. Supplying *E. faecium* to suckling pigs and broiler chickens improved their growth performance and decreased the incidence of diarrhea [39,40]. Supplementation of sow feed with *E. faecium* reduced the weight loss of sow and increased the content of lactose as well as the percentage of milk cells expressing mCD14 in sow milk, which positively correlated to weaning weight and the percentages of B cells and activated T cells in piglets [40]. Moreover, *E. faecium* M-74 was used as the probiotic in the treatment of gastrointestinal complications in patients with myeloid leukemia [41]. Importantly, *E. faecalis* and *E. faecium* also produce metabolites of commercial importance, including organic acids, antimicrobial and aromatic compounds, exopolysaccharides, enterocins, virulence factors, and bioactive peptides, which can facilitate their adhesion and colonization in the digestive system and facilitate adaptation to ecological niches, inhibit the proliferation of spoilage microbes, determine their pathogenicity, stimulate the host immune system, and so on [36,37,38,39,40]. Both *E. faecium* and *E. faecalis*, were widely used as a model for the acquisition of antibiotic resistance in the host [35]. However, the role of these two *Enterococcus* species in transforming bioactive metabolites in food production remains largely unexplored.

Here, we found that *E. faecalis* and *E. faecium* could modify the common versatile precursor metabolite toluquinol by transferring an acetyl group to a OH or a methyl methylsulfonyl unit at C-5 or C-6 of toluquinol, while another two strains, *S. thermophiles* and *L. lactis* subsp., could only transfer a methyl methylsulfonyl unit at C-5 or C-6 of toluquinol. Interestingly, acetylated metabolites **1** and **2** displayed significantly increased inhibitory activity towards A549 compared with toluquinol, suggesting that acetylated metabolites had better antitumor activity than substrate. The enzymes in *Enterococcus* sp. that catalyzed the acetylation of phenol derivatives deserve to be explored and characterized.

Many recent studies suggested that Fenton reaction was responsible for ferroptosis, which was a recently described form of regulated necrosis entirely distinct from other forms of cell death featuring iron-dependent accumulation of lipid peroxidation [42,43]. Through Fenton chemistry, Fe^2+^ combines with peroxides to generate highly reactive hydroxyl radical and peroxyl radicals promoting nonenzymatic oxidation of lipids [42]. Previous study suggested that these reactive hydroxyl radical and peroxyl radicals could turn DMSO into CH_3_ radical and the methylsulfonyl unit and promote a non-enzymatic production of methylsulfonyl benzonquinols [31]. Here, we found that four gut microbes could also take a methylsulfonyl group from DMSO and then transform toluquinol to the methylsulfonyl toluquinols (**3**–**4**). Our finding, for the first time, reported that *E. faecalis, E. faecium, S. thermophiles, and L. lactis* subsp. could also use Fenton chemistry to metabolize the versatile toluquinol (Figure 8).

Fenton’s chemistry in organisms can lead to oxidative stress that could break the cellular homeostasis and result in cellular dysfunctions [42,43]. Many investigations revealed that the oxidative stress in the gastrointestinal tract could lead to various intestinal diseases, such as peptic ulcers, diarrhea, and inflammation [44,45]. Gut microbes play a crucial role in maintaining intestinal redox homeostasis, e.g., *Bacillus* attenuates oxidative stress and could induce intestinal injury via p38-mediated autophagy, and *Lactobacillus plantarum* could alleviate oxidative damage induced by weaning in the lower gut of young goats [46,47]. Our results suggested that *E. faecalis*, *E. faecium*, *S. thermophiles*, and *L. lactis subsp*. might have important application values in maintaining redox homeostasis and treating diseases caused by intestinal redox imbalance.

## 3. Materials and Methods

### 3.1. Penicillium Fungal Strain and Cultivation

A fungal strain was collected and purified from Chinese steamed bread stored in a fridge for two weeks and infected with cyan *Penicillium* species. Sequencing analysis showed that the fungus shared 100% identity of ITS sequences to *Penicillium rubens*. *Penicillium rubens* CSB10001 targeted ITS sequences for this study have been deposited in the European Nucleotide Archive (ENA) at EMBL-EBI (Wellcome Genome Campus, Hinxton, Cambridgeshire, UK) under accession number PRJEB53890. The fungal strain was stored in the State Key Laboratory for Conservation and Utilization of Bio-Resources & Key Laboratory for Microbial Resources of the Ministry of Education. PDA (potato (Kunming, Yunnan, China) 200 g/L, glucose () 10 g/L, agar () 15 g/L) medium was used for analyzing mycelial growth and related phenotypic traits. The fungal strain was cultured on PDA medium at room temperature to obtain conidia, and then the conidia were inoculated in some Chinese steamed breads (total 100 g) and some tangerines (total 200 g), respectively. The inoculated foods were left at room temperature for 4–6 days.

### 3.2. Gut Microbial Strains and Cultivation

Seventeen strains belonging to 12 genera, as listed in Table 1, were collected from the State Key Laboratory for Conservation and Utilization of Bio-Resources & Key Laboratory for Microbial Resources of the Ministry of Education. The 17 gut microbial strains were firstly cultivated in a suitable solid media and then cultured in the corresponding liquid media, as described in Table 2. MRS medium was prepared with proteose peptone (Macklin, Shanghai, China) 10 g/L, beef extract 10 g/L, yeast extract 5 g/L, dextrose (Sigma-Aldrich, MA, USA) 20 g/L, polysorbate 80 (Aladdin, Shanghai, China) 1 g/L, ammonium citrate 2 g/L, sodium acetate 5 g/L, magnesium sulphate 0.1 g/L, manganese sulphate 0.05 g/L, and dipotassium phosphate 2g/L. YM medium was prepared with yeast extract 3 g/L, malt extract 3 g/L, peptone 5 g/L, and dextrose 10 g/L. Lactic acid bacteria medium (LAB) was prepared with yeast extract 7.5 g/L, glucose 10.0 g/L, tomato juice 100 mL/L, peptone 7.5 g/L, potassium dihydrogen phosphate 2.0 g/L, and Tween 80 0.5 mL/L. TSA medium was prepared with soy peptone 5.0 g/L, tryptone 15.0 g/L, and sodium chloride 5.0 g/L. The solid TSA medium (blood plate medium) contains TSA and 5% defibrinated sheep blood. Nutrient agar medium (NA) was prepared with beef extract 3 g/L, peptone 5 g/L, and NaCl 10 g/L. The culturing temperatures and media for these 17 gut microbial strains are described in Table 1.

### 3.3. LC–MS Analysis

Metabolic analysis was performed by ultraperformance liquid chromatography with diode array detection MS (UPLC-DAD/MS). The fermentation broths and foods were extracted with the same volume of ethyl acetate overnight, and the organic layers were evaporated and dissolved in 1 mL of menthol and filtered through a 0.22 μm filter membrane. All of the samples were performed by a UPLC system coupled to a Q Exactive Focus Orbitrap mass spectrometer (Thermo Fisher, Waltham, MA, USA) equipped with an Agilent Zorbax ODS 4.6 by 250 mm column (Agilent, Santa Clara, CA, USA) using the electrospray ionization (ESI) mode. The data were analyzed by Xcalibur software.

### 3.4. Isolation of Targeted Metabolites in E. faecalis or Enterococcus faecium

The 50 L cultural broths of *E. faecalis* or *Enterococcus faecium* cultured on MRS medium with 200 µM toluquinol were concentrated in vacuo and partitioned with acetyl acetate. The acetyl acetate parts were evaporated to dryness to give residue. The crude extract of the culture was subjected to purification first with absorption resin D101, then by reversed-phase (RP18) flash chromatography eluting with H_2_O–MeOH (10–90%) with decreasing polarity through a continuous gradient, and subsequently by open-column chromatography on Sephadex LH-20 with acetone, respectively. Silica gel 60 (Merck, 200–400 mesh) was used for column chromatography. Column chromatography was performed on 200–300 mesh silica gel (Qingdao Marine Chemical Factory, Qingdao, Shandong, China). The TLC spots were detected by spraying the TLC plates with 20% (*w*/*v*) H_2_SO_4_ and then heating them on a hot plate. The purified samples were structurally elucidated with NMR and MS data.

### 3.5. General Spectra for Structural Characterization

The purified metabolites were analyzed with NMR and MS. NMR experiments were carried out on a Bruker DRX-500 spectrometer (Bruker, Madison, WI, USA) with solvent as internal standard. MS data were recorded on a VG-Auto-Spec-3000 spectrometer (VG, Manchester, UK). High-resolution ESI MS data were measured on a Bruker Bio-TOF III electrospray ionization mass spectrometer. X-ray diffraction was realized on a Bruker APEX DUO crystallography system (Bruker, Madison, WI, USA).

4-O-acetyltoluquinol (**1**): C_9_H_10_O_3_; Colorless oil; UV (MeOH) λmax (log *ε*) 194, 278; ^1^H NMR (CDCl_3_, 500 MHz) *δ*_H_ 6.77 (d, *J* = 2.8 Hz, 1H, H-3), 6.69 (dd, *J* = 8.6, 2.8 Hz, 1H, H-5), 6.63 (d, *J* = 8.6 Hz, 1H, H-6), 2.14 (s, 3H, 2-Me), 2.20 (s, 3H, Ac-Me); Negative ESI-MS *m*/*z* 165 [M–H]^−^.

1-O-acetyltoluquinol (**2**): C_9_H_10_O_3_; Colorless oil; UV (MeOH) λmax (log *ε*) 194, 278; ^1^H NMR (CDCl_3_, 500 MHz) *δ*_H_ 6.55 (d, *J* = 3.0 Hz, 1H, H-3), 6.50 (dd, *J* = 8.6, 3.0 Hz, 1H, H-5), 6.74 (d, *J* = 8.6 Hz, 1H, H-6), 2.03 (s, 3H, 2-Me), 2.22 (s, 3H, Ac-Me); ^13^C NMR(CDCl_3_, 125 MHz) *δ*_C_ 142.7 (s, C-1), 131.2 (s, C-2), 117.6 (d, C-3), 153.5 (s, C-4), 113.6 (d, C-5), 122.5 (d, C-6), 16.2 (q, 2-Me-C), 171.9 (s, Ac-C=O), 20.8 (q, Ac-Me); Negative ESI-MS *m*/*z* 165 [M–H]^−^.

5-Methylsulfonyl toluquinol (**3**): C_8_H_10_O_4_S; Colorless oil; UV (MeOH) λmax (log *ε*) 199, 307; ^1^H NMR (CDCl_3_, 500 MHz) *δ*_H_ 7.08 (s, 1H, H-3), 6.65 (s, 1H, H-6), 2.15 (s, 3H, 2-Me), 3.32 (s, 3H, sulfone-Me); ^13^C NMR (CDCl_3_, 125 MHz) *δ*_C_ 149.7 (s, C-1), 130.7 (s, C-2), 117.1 (d, C-3), 145.6 (s, C-4), 136.3 (s, C-5), 115.1 (d, C-6), 15.4 (q, 2-Me-C), 43.1 (s, Sulfone-Me); Negative HRESI-MS *m*/*z*: 201.02170 [M–H]^−^, calculated as 201.0221 for C_8_H_9_O_4_S. Crystal data for **3**: C_8_H_10_O_4_S, *M* = 202.22, *a* = 11.0168(19) Å, *b* = 5.8970(10) Å, *c* = 13.201(2) Å, *α* = 90°, *β* = 92.500(6)°, *γ* = 90°, *V* = 856.8(3) Å^3^, *T* = 100(2) K, space group *P*121/*n*1, *Z* = 4, *μ*(Cu Kα) = 3.227 mm^−1^, 10632 reflections measured, 1614 independent reflections (*R_int_* = 0.0750). The final *R*_1_ values were 0.0643 (*I* > 2*σ*(*I*)). The final *wR*(*F*^2^) values were 0.1849 (*I* > 2*σ*(*I*)). The final *R*_1_ values were 0.0714 (all data). The final *wR*(*F*^2^) values were 0.1946 (all data). The goodness of fit on *F*^2^ was 1.133.

6-Methylsulfonyl toluquinol (**4**): C_8_H_10_O_4_S; Colorless oil; UV (MeOH) λmax (log *ε*) 210, 308; Negative HRESI-MS *m*/*z*: 201.02184 [M–H]^−^, calculated as 201.0221 for C_8_H_9_O_4_S.

### 3.6. Cell Culture and Cell Viability Assay

Human cancer cell lines were purchased from the Kunming Cell Bank of the Chinese Academy of Science. HCT116, A2780, G401, LM3, and A375 cells were all cultured in DMEM (Gibco, Carlsbad, CA, USA) supplemented with 10% fetal bovine serum (BI, Kibbutz Beit Haemek, Israel) and 1% penicillin/streptomycin (Gibco); A549 cells were cultivated in RPMI-1640 medium (Gibco). All cells were cultured at 37 °C in a 5% CO_2_ humidified environment. For the cell viability assays, the cells were plated at a density of 5000 cells/well in 96-well plates and incubated overnight. The cells were then treated with CCK-8 for 72 h and assessed using a microplate reader at 450 nm.

### 3.7. Flow Cytometry Analysis

A549 cells cultured in RPMI-1640 medium were treated with 2 µM **1**, **2,** and **3** compounds for 24 h, respectively. For cell cycle analysis, harvested cells were washed twice in PBS and fixed in ice-cold 75% ethanol for 48 h. The cells were then washed twice with cold PBS and incubated in PBS mixing with 50 µg/mL propidium iodide (Sigma, St. Louis, MO, USA) and 20 µg/mL RNase A for 30 min at 37 °C. For the analysis of apoptosis and necrosis, the Apoptosis and Necrosis Detection Kit with YO-PRO-1 and PI (Beyotime, Nantong, China) were used according to the manufacturer’s instructions. The samples were tested on a flow cytometer (BD, LSR Fortessa, San Jose, CA, USA) and the data were analyzed using FlowJo VX software.

### 3.8. Statistical Analysis

All cellular experiments were repeated at least three times. Data were presented as means ± standard deviation (SD). GraphPad 9.0 software was used for statistical analysis. Comparisons were performed using the two-tailed Student’s *t*-test. *p* < 0.05 was used as the threshold for determining significant differences (* *p* < 0.05; ** *p* < 0.01; *** *p* < 0.001).

## Figures and Tables

**Figure 1 ijms-23-10700-f001:**
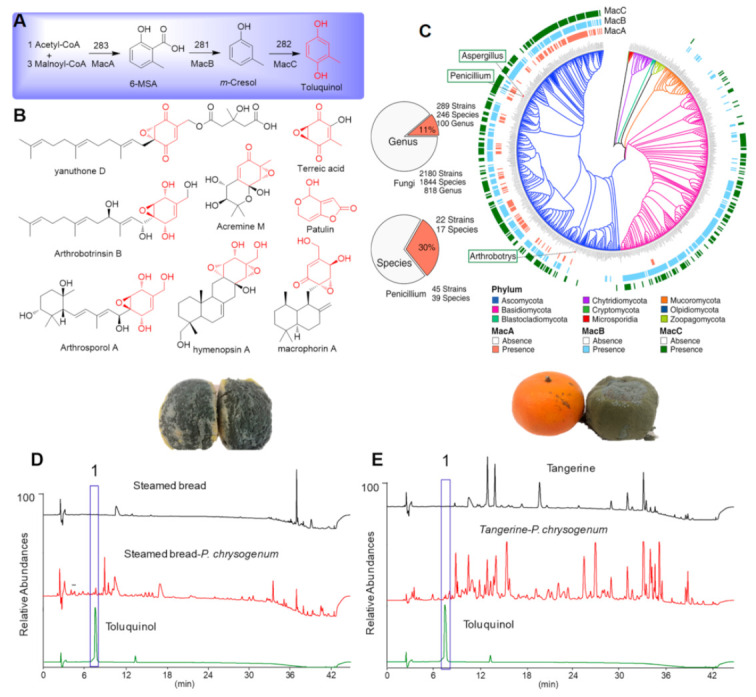
Biosynthesis and wide distribution of toluquinol as a key intermediate for a large number of fungal metabolites. (**A**) Biosynthetic pathway for toluquinol in fungi. (**B**) Structures of representative fungal metabolites using toluquinol as a key precursor. (**C**) Wide distribution of the key biosynthetic gene *PKS* for toluquinol in fungi. (**D**) Detection of toluquinol in the metabolic profile of Chinese steamed breads colonized with *Penicillus* species. (**E**) Detection of toluquinol in the metabolic profile of Tangerines colonized by *Penicillus* species.

**Figure 2 ijms-23-10700-f002:**
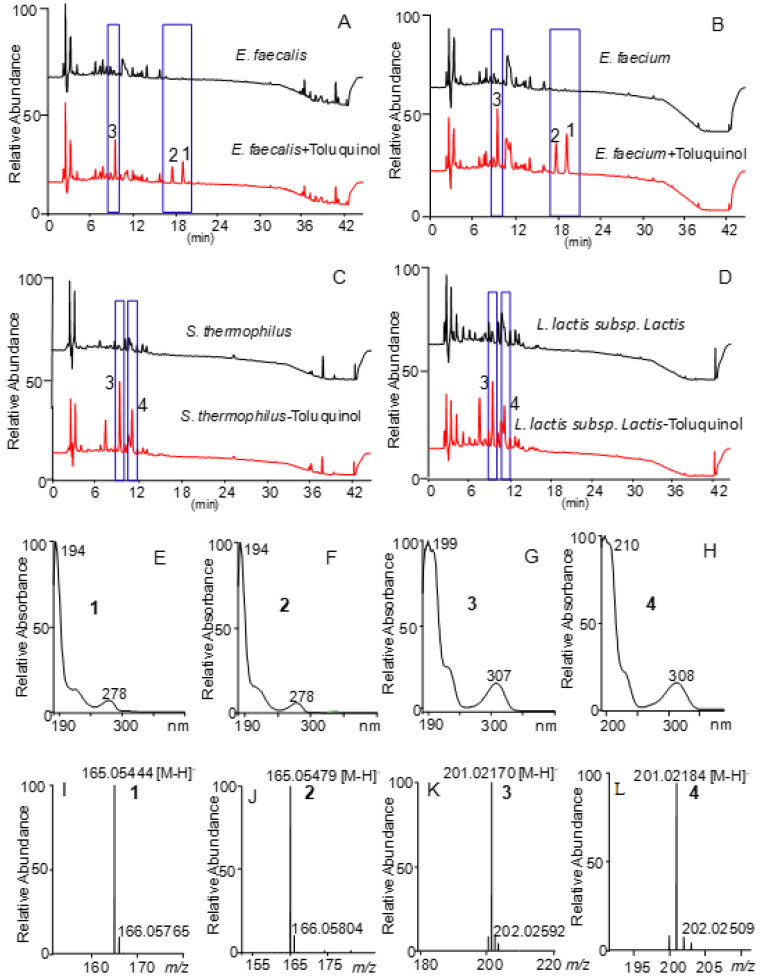
Transformation of toluquinol in four gut microbes. (**A**–**D**) Comparison of metabolic profiles of *E. faecalis*, *E. faecium*, *S. thermophilus*, and *L. lactis* subsp. *Lactis* fed with and without toluquinol. (**E**–**H**) UV spectra of the targeted metabolites **1**–**4**. (**I**–**L**) Mass spectra of the targeted metabolites **1**–**4**.

**Figure 3 ijms-23-10700-f003:**
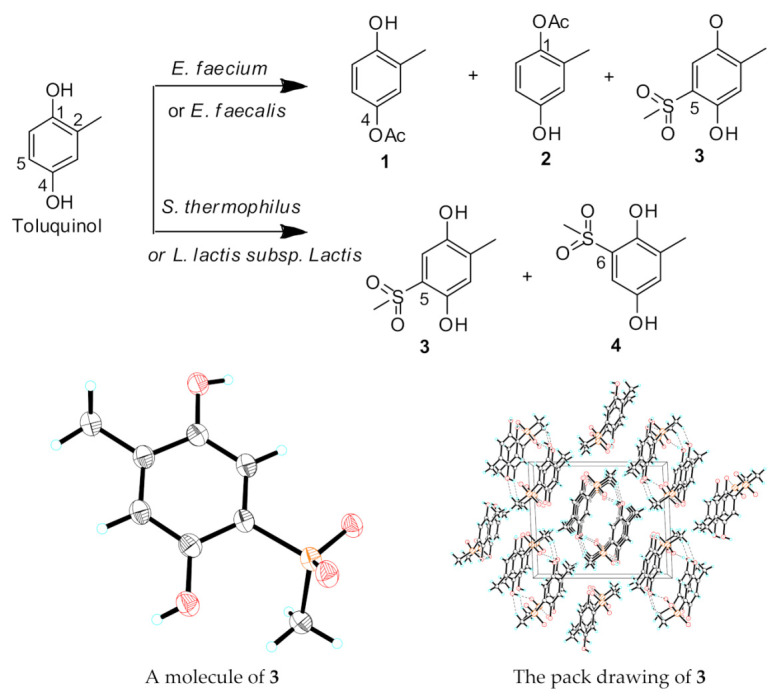
The four targeted metabolites (**1**–**4**) derived from toluquinol in four gut microbes and the single X–ray crystallographic structure of **3**. Red: oxygen; Orange: Sulfur; Blue: Hydrogen; Black: Carbon.

**Figure 4 ijms-23-10700-f004:**
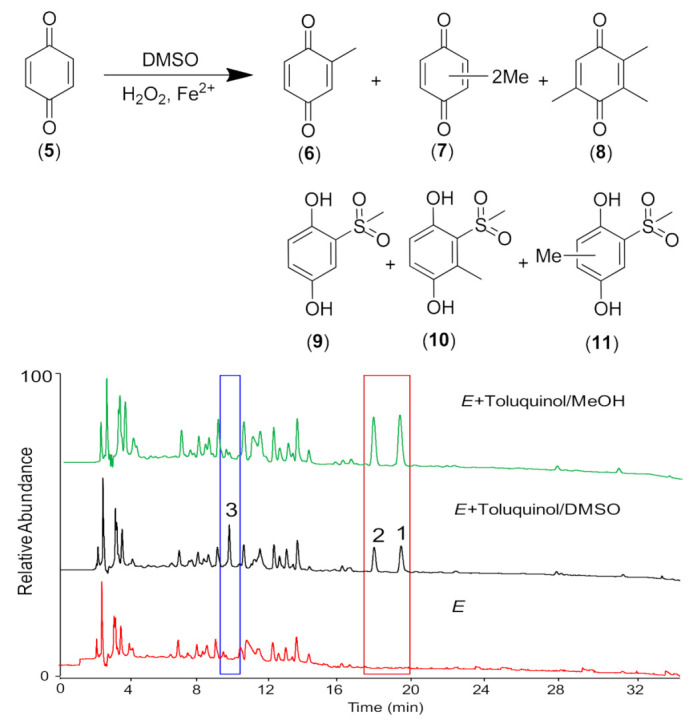
The artificial products (**6**–**11**) from benzonquinone (**5**) and DMSO via Fenton’ reaction and comparison of metabolic profiles of *E. faecalis* (*E*) fed with toluquinol in DMSO and MeOH.

**Figure 5 ijms-23-10700-f005:**
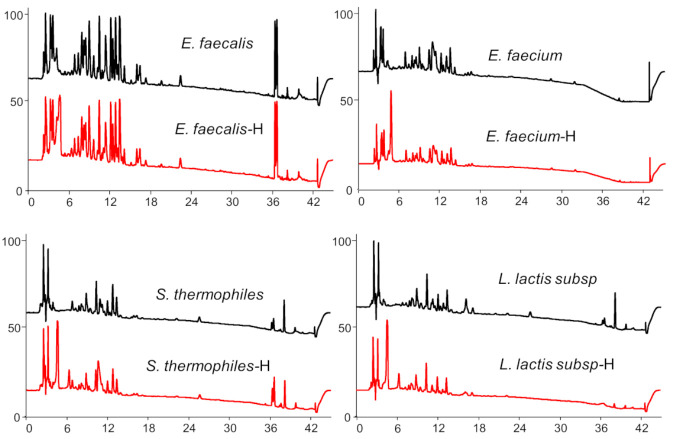
Comparison of metabolic profiles of four strains, *E. faecalis*, *E. faecium*, *S. thermophilus*, and *L. lactis* subsp. *Lactis*, fed with and without hydroquinone (H, **5**) in DMSO.

**Figure 6 ijms-23-10700-f006:**
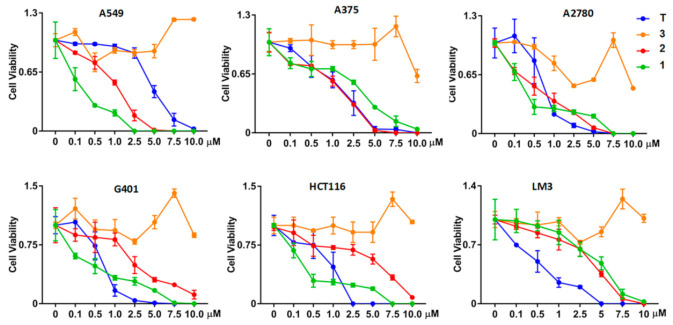
Effects of toluquinol-derived metabolites **1**–**3** and toluquinol on human tumor cell lines including non-small-cell lung cancer cell A549, skin melanoma cell A375, ovarian cancer cell A2780, kidney tumor cell G401, colorectal cancer cell HCT116, and hepatocellular carcinoma LM3.

**Figure 7 ijms-23-10700-f007:**
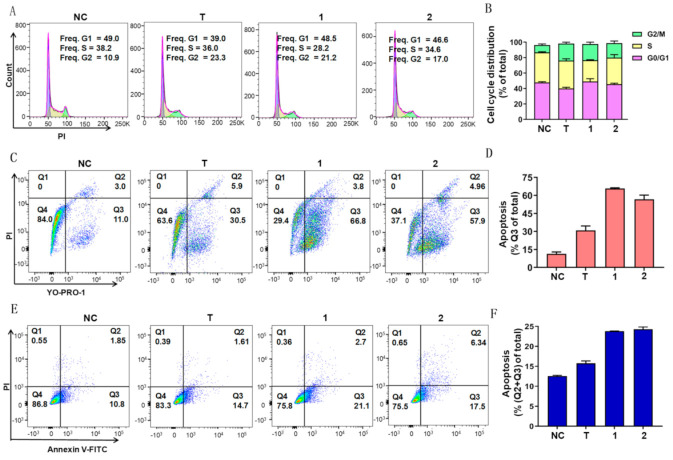
Effects of toluquinol and its derived metabolites **1**–**2** on cell cycle, apoptosis, and necrosis of non-small lung cancer cell A549. (**A**,**B**) Analysis of A549 cell cycle distributions with flow cytometry and PI staining. (**C**,**D**) Apoptosis and necrosis analysis of A549 with flow cytometry and YO-PRO-1/PI staining. The apoptotic cells were YO-PRO-1 positive (gate Q3), and the necrotic cells were both YO-PRO-1 and PI positive (gate Q2). (**E**,**F**) Apoptosis analysis of A549 with AnnexinV-FITC/PI staining. The early apoptotic cells were AnnexinV-FITC positive (gate Q3), and late apoptotic cells were both AnnexinV-FITC and PI positive (gate Q2). Negative control: NC; T: Toluquinol.

**Figure 8 ijms-23-10700-f008:**
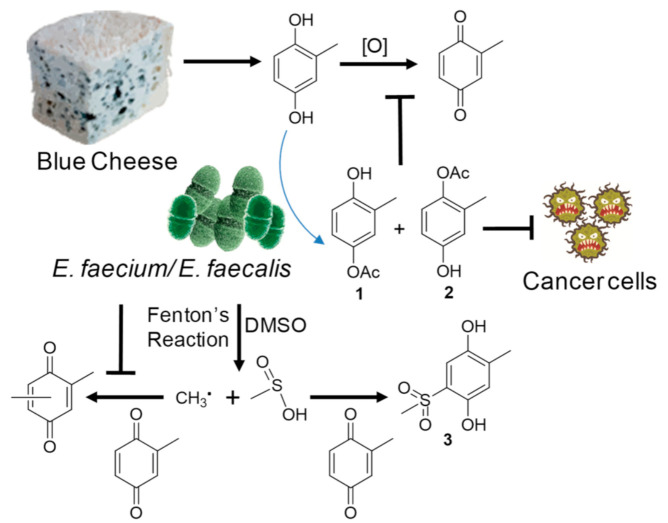
Effects of *E. faecalis* and *E. faecium* on transforming versatile toluquinol from *Penicillum* species.

**Table 1 ijms-23-10700-t001:** IC_50_ values for toluquinol-derived metabolites **1**–**3** towards human tumor cell lines.

	IC_50_ (μM)					
Cell	A549	A375	A2780	G401	HCT116	LM3
T	4.781	1.716	0.687	0.602	0.91	0.613
**1**	0.224	4.175	0.204	0.597	3.767	4.993
**2**	1.275	1.951	0.741	3.412	7.0255	4.23
**3**	-	-	-	-	-	-

**Table 2 ijms-23-10700-t002:** The list for gut microbe strains and their cultural temperatures (T) and media (M).

No.	Strains	T (°C)	M
1	*Bacillus coagulans* GDMCC1.646	37	MRS
2	*Bacillus licheniformis* BNCC132620	37	NA
3	*Bacillus subtilis* BNCC188062	30	NA
4	*Enterococcus faecalis* BNCC194769	30	MRS
5	*Enterococcus faecium* BNCC194768	30	MRS
6	*Lactobacillus bulgaricus* GDMCC1.189	37	MRS
7	*Lactococcus lactis* subsp. *Lactis* ATCC11007	37	LAB
8	*Lactobacillus planturum* BNCC336421	37	MRS
9	*Lactobacillus salivarius* BNCC194720	37	MRS
10	*Leuconstoc meseteroides* GDMCC1.774	37	MRS
11	*Kluyveromyces marx* ATCC36534	25~28	YM
12	*Pediococcus acidilactici* GDMCC1.263	37	MRS
13	*Pediococcus pentosaceus* BNCC193259	37	MRS
14	*Saccharomyces cerevisiae* BNCC336503	28~30	YM
15	*Staphylococcus vitulinus* DMS15615	37	TSA
16	*Staphylococcus xylosus* BNCC37469	30	NA
17	*Stretpococcus thermophilus* GDMCC1.1808	37	LAB

## Data Availability

Additional supporting information may be found in the online version of this article on the publisher’s website.

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
