# Peer review of "Modulating Activity Evaluation of Gut Microbiota with Versatile Toluquinol"

_ijms, 2022, doi:10.3390/ijms231810700_

Round 1

Reviewer 1 Report

In this experimental study, fungal metabolites, i.e., toluquinol-derived compounds, were evaluated for their biological activities towards different cancer cells.

An introduction section describes the role of gut microbiota in human health, emphasizing that the role of specific gut microbial strains in modulating specific intermediate metabolites from fungal origins in the diet is still lacking.

Sentences in lines 55-58 are unclear.

In lines 93-96, the authors mentioned 17 and 34 microbial strains. Then, 17 common strains of human intestinal probiotics belonging to 12 genera were fermented and fed with toluquinol. All the 17 strains fed with solvent only were used as negative controls. The metabolic profiles of the extracts of all the 34 gut microbial strains were analyzed with HPLC-MS/PDA.”

Why 34 strains if you have analyzed 17? Clarify this!

In the section on Results and Discussion, Figures 1D and 1E do not correspond with the text in paragraph 2.2. (check Penicillium species)!  

Check Figures S1-S2 legends.

The methodology section lacks data for the proteose peptone amount (line 337).

This study provides valuable details regarding microbial strains and their effects on fungal metabolites.

The paper complies with the field of this journal.

Author Response

Dear Reviewer,

Thank you very much for taking time out of your busy schedule to read this article. Thank for providing the constructive modification comments and suggestions, which have greatly helped us to further improve this manuscript. We carefully thought and analyzed your valuable opinions. On this basis, we have made the appropriate revisions and responses to your comments and suggestions, as noted below.

Sentences in lines 55-58 are unclear.

Response: Thanks for nice suggestion. The sentence “Previous studies revealed that a polyketide synthase gene cluster (PKS) that was responsible for the biosynthesis of the famous patulin.” has been changed to “Previous studies revealed that a polyketide synthase gene cluster (PKS) containing a 6-methyl salicylic acid synthase gene was responsible for the biosynthesis of the well-known patulin.”

In lines 93-96, the authors mentioned 17 and 34 microbial strains. “Then, 17 common strains of human intestinal probiotics belonging to 12 genera were fermented and fed with toluquinol. All the 17 strains fed with solvent only were used as negative controls. The metabolic profiles of the extracts of all the 34 gut microbial strains were analyzed with HPLC-MS/PDA.” Why 34 strains if you have analyzed 17? Clarify this!

Response: Thanks for nice suggestion. It was a typo. To evaluate the effects of 17 different gut microbial strains on the transformation of toluquinol, these 17 strains fed with and without toluquinol, respectively. Thus, there were 34 gut microbial samples for HPLC analysis. “The 34 gut microbial strains” has been changed to “34 gut microbial samples”.

In the section on Results and Discussion, Figures 1D and 1E do not correspond with the text in paragraph 2.2. (check Penicillium species)!  

Response: Thanks for nice suggestion. The Penicillium species in the text in paragraph 2.2. have been corrected to Penicillium chrysogenum which was described in Figures 1D and 1E.

Check Figures S1-S2 legends.

Response: Figure S1 legend has been changed to “Comparison of metabolic profiles of the 13 gut microbes fed with and without toluquinol suggested no transformation of toluquinol in 13 strains including Bacillus coagulans Bacillus licheniformis, Bacillus subtilis, Kluyveromyces marx, Leuconstoc meseteroides, Pediococcus acidilactici, Saccharomyces cerevisiae, Lactobacillus bulgaricus, Lactococcus lactis subsp. Lactis, Lactobacillus planturum, Lactobacillus salivarius, Pediococcus pentosaceus, Staphylococcus vitulinus, Staphylococcus xylosus, and Stretpococcus thermophilus.”

Figure S2 legend has been changed to “Comparison of metabolic profiles of 12 gut microbes fed with and without hydroquinone (H) suggested no transformation of hydroquinone in these 12 strains including Bacillus coagulans Bacillus licheniformis, Bacillus subtilis, Kluyveromyces marx, Leuconstoc meseteroidesPediococcus acidilactici, Saccharomyces cerevisiae, Lactobacillus bulgaricus, Lactococcus lactis subsp. Lactis, Lactobacillus planturum, Lactobacillus salivarius , Pediococcus pentosaceus, Staphylococcus vitulinus, and Staphylococcus xylosus.

The methodology section lacks data for the proteose peptone amount (line 337).

Response: The amount data 10 g/L for the proteose peptone was added in the section.

Thanks for your great efforts.

Best Wishes!

Xuemei

Reviewer 2 Report

Review comments

The research article by Long-Long Zhang et al titled “Modulating Activity Evaluation of Gut Microbiota with Versa-2 tile Toluquinol” is a very interesting study. Some of my Major concerns are:

1.     In Figure 7A, data show that cell number in G0/G1 phase is higher compared to other cell cycle phases. However, sub G1 phase cell number among treatments groups is not shown. Most studies have shown a peak in subG1 phase of cell cycle when cells undergo apoptotic cell death (a).

Reference

(a)   Wang YY, Kwak JH, Lee KT, Deyou T, Jang YP, Choi JH. Isoflavones Isolated from the Seeds of Millettia ferruginea Induced Apoptotic Cell Death in Human Ovarian Cancer Cells. Molecules. 2020 Jan 3;25(1):207. doi: 10.3390/molecules25010207. PMID: 31947862; PMCID: PMC6983189.

2.     To strengthen the study and confirmation of apoptosis in western blotting, it is always good to show the pro and cleaved form of caspase 3 and caspase 9 to show actual caspase activity. In addition, authors can also show its downstream, total and cleaved form of PARP to prove the execution of apoptosis after cadmium treatment. If needed but not so required as the authors clearly showed through Flow cytometry data which is very convincing.

3.      The authors have clearly shown NMR, MS and X-ray diffraction analysis. I recommend acceptance of the manuscript after some small corrections. 

Author Response

Dear Reviewer,

Thank you very much for taking time out of your busy schedule to read this article. Thank for providing the constructive modification comments and suggestions, which have greatly helped us to further improve this manuscript. We carefully thought and analyzed your valuable opinions. On this basis, we have made the appropriate revisions and responses to your comments and suggestions, as noted below.

  1. In Figure 7A, data show that cell number in G0/G1 phase is higher compared to other cell cycle phases. However, sub G1 phase cell number among treatments groups is not shown. Most studies have shown a peak in subG1 phase of cell cycle when cells undergo apoptotic cell death (a).

Reference:(a)   Wang YY, Kwak JH, Lee KT, Deyou T, Jang YP, Choi JH. Isoflavones Isolated from the Seeds of Millettia ferruginea Induced Apoptotic Cell Death in Human Ovarian Cancer Cells. Molecules. 2020 Jan 3;25(1):207. doi: 10.3390/molecules25010207. PMID: 31947862; PMCID: PMC6983189.

Response: Thanks for your suggestion. Indeed, some papers showed that the subG1 phase of cell cycle could be used as an indicator for apoptosis. However, there were some papers which suggested that the apoptotic cells could not be detected in cell cycle analysis (please referring to the following references 1-3). Thus, another apoptosis assay was performed to evaluate the cell death. In our study, the cell cycle analysis was carried out after A549 cells were treated with T, 1 or 2 for 24 h, while the apoptosis analysis was performed after A549 cells were treated for 48 h. According to the results from two assays, we deduced that the apoptosis induced by T, 1 or 2 after arresting the cell cycle of A549 at G2/M phase.

1) Auti, A., Alessio, N., Ballini, A., Dioguardi, M., Cantore, S., Scacco, S., Vitiello, A., Quagliuolo, L., Rinaldi, B., Santacroce, L., Di Domenico, M., & Boccellino, M. (2022). Protective Effect of Resveratrol against Hypoxia-Induced Neural Oxidative Stress. Journal of personalized medicine, 12(8), 1202. https://doi.org/10.3390/jpm12081202

2) Wang, J., Wang, L., Zhang, S., Fan, J., Yang, H., Li, Q., & Guo, C. (2020). Novel eIF4E/eIF4G protein-protein interaction inhibitors DDH-1 exhibits anti-cancer activity in vivo and in vitro. International journal of biological macromolecules, 160, 496–505. https://doi.org/10.1016/j.ijbiomac.2020.05.233

3) Maggi, F., Morelli, M. B., Tomassoni, D., Marinelli, O., Aguzzi, C., Zeppa, L., Nabissi, M., Santoni, G., & Amantini, C. (2022). The effects of cannabidiol via TRPV2 channel in chronic myeloid leukemia cells and its combination with imatinib. Cancer science, 113(4), 1235–1249. https://doi.org/10.1111/cas.15257

  1. To strengthen the study and confirmation of apoptosis in western blotting, it is always good to show the pro and cleaved form of caspase 3 and caspase 9 to show actual caspase activity. In addition, authors can also show its downstream, total and cleaved form of PARP to prove the execution of apoptosis after cadmium treatment. If needed but not so required as the authors clearly showed through Flow cytometry data which is very convincing.

Response: Thanks for your nice suggestion. The analysis of the pro and cleaved form of caspase 3 and caspase 9 and the total and cleaved form of PARP can provide another evidence for the apoptosis. However, it will take more two weeks to prepare and carry out these experiments. We are so sorry that we cannot afford it in five days. To provide more solid data for the apoptosis in five days, another assay for the apoptosis by using the AnnexinV-FITC/PI staining analysis were carried out (Figure 7E-7F in the revised manuscript, please see the following Figure). The total apoptosis (Q2+Q3) rates in the T, 1 and 2 groups were much higher than that in NC groups. The result was totally consistent with the result from the apoptosis analysis with YO-PRO-1/PI staining. We hope that the added data could also confirm the apoptosis as well.

Figure 7. Effects of toluquinol, and its derived metabolites 12 on cell cycle, apoptosis and necrosis of non-small lung cancer cell A549. (A–B) Analysis of A549 cell cycle distributions with flow cytometry and PI staining. (C–D) Apoptosis and necrosis analysis of A549 with flow cytometry and YO-PRO-1/PI staining. The apoptotic cells were YO-PRO-1 positive (gate Q3), and necrotic cells were both YO-PRO-1 and PI positive (gate Q2). (E–F) Apoptosis analysis of A549 with AnnexinV-FITC/PI staining. The early apoptotic cells were AnnexinV-FITC positive (gate Q3), and late apoptotic cells were both AnnexinV-FITC and PI positive (gate Q2). Negative control: NC; T: Toluquinol.

Thanks for your great efforts.

Best wishes!

Xuemei